Citation: *Molecular Systems Biology* 9:683
www.molecularsystemsbiology.com

# Bacterial cheating drives the population dynamics of cooperative antibiotic resistance plasmids

Eugene A Yurtsev[1,3], Hui Xiao Chao[1,3], Manoshi S Datta[2], Tatiana Artemova[1] and Jeff Gore[1,*]

[1] Department of Physics, Massachusetts Institute of Technology, Cambridge, MA, USA and [2] Computational and Systems Biology Program, Massachusetts Institute of Technology, Cambridge, MA, USA
[3]These authors contributed equally to this work.
* Corresponding author. Department of Physics, Massachusetts Institute of Technology, 77 Massachusetts Avenue, Cambridge, MA 02139, USA.
Tel.: + 1 617 715 4251; Fax: + 1 617 258 6883; E-mail: gore@mit.edu

Inactivation of β-lactam antibiotics by resistant bacteria is a 'cooperative' behavior that may allow sensitive bacteria to survive antibiotic treatment. However, the factors that determine the fraction of resistant cells in the bacterial population remain unclear, indicating a fundamental gap in our understanding of how antibiotic resistance evolves. Here, we experimentally track the spread of a plasmid that encodes a β-lactamase enzyme through the bacterial population. We find that independent of the initial fraction of resistant cells, the population settles to an equilibrium fraction proportional to the antibiotic concentration divided by the cell density. A simple model explains this behavior, successfully predicting a data collapse over two orders of magnitude in antibiotic concentration. This model also successfully predicts that adding a commonly used β-lactamase inhibitor will lead to the spread of resistance, highlighting the need to incorporate social dynamics into the study of antibiotic resistance.

*Molecular Systems Biology* **9**: 683; published online 6 August 2013; doi:10.1038/msb.2013.39
*Subject Categories:* simulation and data analysis; microbiology & pathogens
*Keywords:* antibiotic inactivation; antibiotic resistance; cooperation and cheating; β-lactam; population dynamics

## Introduction

A frequent mechanism of antibiotic resistance involves the production of an enzyme that inactivates the antibiotic (Davies, 1994; Wright, 2005). The acquisition of such an enzyme through a plasmid often imposes a metabolic cost on the individual cell (Bouma and Lenski, 1988; Dahlberg and Chao, 2003; Andersson, 2006); however, since resistant cells inactivate the antibiotic, reducing its extracellular concentration, they help protect the entire bacterial population (Dugatkin *et al*, 2003; Brook, 2004). Hence, antibiotic inactivation can be viewed as a cooperative behavior, suggesting that sensitive 'cheater' bacteria that do not help to break down the antibiotic may be able to survive antibiotic treatment in the presence of resistant cells.

Previous studies have provided valuable insight into the evolutionary processes that govern the spread of antibiotic resistance (Neu, 1992; Goossens *et al*, 2005; Weinreich *et al*, 2006; Lee *et al*, 2010; Zhang *et al*, 2011; Toprak *et al*, 2012). However, despite the clinical importance of antibiotic resistance phenotypes, there has been a relative dearth of quantitative analysis of cooperative bacterial growth in the presence of antibiotics. Many microbiologists have observed the presence of 'satellite colonies' surrounding a resistant colony on an agar plate containing the β-lactam ampicillin.

The presence of satellite colonies, which are composed of cells that are in principle unable to grow in ampicillin, is evidence of the extremely cooperative nature of ampicillin resistance. Indeed, recent experiments have detected coexistence between resistant and sensitive cells using a resistance enzyme that was genetically modified to inactivate the antibiotic outside the cell (Dugatkin *et al*, 2005; Perlin *et al*, 2009). Furthermore, it is known in the clinic that bacteria carrying even wild-type enzymes may provide protection to pathogenic but otherwise sensitive bacteria (Hackman and Wilkins, 1975; Brook, 1984, 2004). The ability of sensitive bacteria to survive antibiotic treatment suggests that the spread of plasmids that encode cooperative antibiotic resistance genes should exhibit non-trivial population dynamics.

## Results

### Population dynamics of antibiotic resistance plasmids

To probe the population dynamics of such plasmids, we co-cultured a sensitive strain of *E. coli* bacteria with an isogenic strain containing an additional plasmid encoding a β-lactamase enzyme. The enzyme hydrolytically inactivates the antibiotic (Bonomo and Tolmasky, 2007), providing

high-level resistance against ampicillin. In our experiments, the bacterial culture was grown to saturation over 23 h in the presence of ampicillin. The saturated culture was then diluted (initially by 100 ×) into fresh media containing the same initial antibiotic concentration, serving as the starting culture for the following day. Using flow cytometry, we were able to track how the fraction of resistant cells changed over time (Materials and methods; Supplementary Figures S1 and S2).

We found that in the presence of resistant bacteria, sensitive bacteria survived and even thrived at a clinically relevant (Foulds, 1986) antibiotic concentration of 100 µg/ml, which is 50-fold larger than their minimum inhibitory concentration (MIC) (Figure 1A; Supplementary Figure S3). A bacterial population with a high fraction of resistant cells inactivated the antibiotic quickly, allowing its sensitive cells to increase in frequency. Over time, the resistant fraction decreased until finally settling to a value of ~0.25. To test whether this fraction corresponded to an equilibrium fraction, we started a

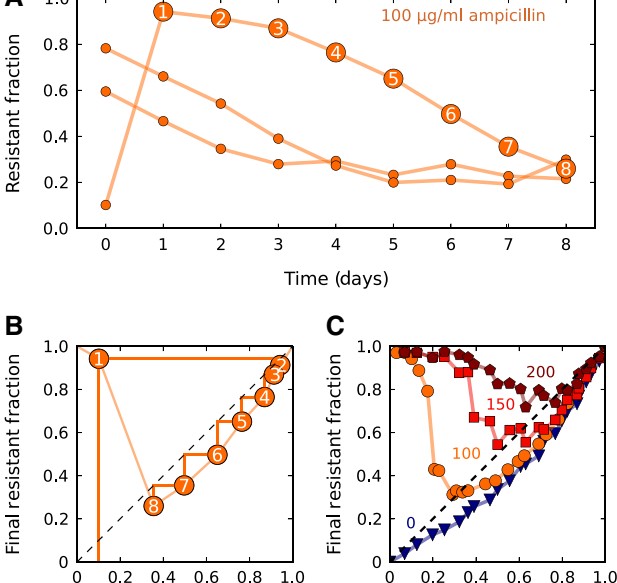

**Figure 1** In the presence of resistant cells, sensitive cells can survive at otherwise lethal antibiotic concentrations. (**A**) Experimental time traces showing the evolutionary dynamics between sensitive *E. coli* and an isogenic strain that is resistant as the result of a plasmid containing a β-lactamase gene. A single resistant and a single sensitive colony were used to create three cultures with a different initial fraction of resistant cells. These three cultures were then grown for 1 day in the absence of ampicillin to make sure that resistant and sensitive cells experienced the same growth conditions (see Materials and methods). Then, every 23 h, the fraction of resistant cells was measured using flow cytometry, and the cultures were diluted by a factor of 100 × into fresh media containing 100 µg/ml ampicillin. Each data point represents a single flow cytometry measurement. (**B**) The orange time trace that starts at ~10% in subplot (A) was replotted as a difference equation map that shows how the resistant fraction on day $n+1$ depends on the fraction on day $n$. The light orange line is an estimation of the difference equation. A simple trick to estimate the time dynamics with a difference equation is to use cobwebbing (dark orange lines), in which the daily dynamics are obtained by bouncing back and forth between the data line and the dashed diagonal line. (**C**) For each antibiotic concentration (indicated adjacent to each curve), a difference equation map was obtained experimentally by starting populations at 24 different initial fractions and measuring the final fraction after 23 h of growth. The intersection of a given difference equation map with the diagonal line represents the equilibrium fraction for that particular condition.

culture at a fraction below the supposed equilibrium. One might have expected the resistant fraction to gradually converge to the equilibrium value. Instead, the resistant fraction initially overshot the equilibrium, jumping to ~0.95, and only then proceeded to decay to the equilibrium. The resistant fraction at the end of the day therefore depends non-monotonically on the resistant fraction at the beginning of the day.

## Using difference equation maps to study population dynamics

Since the final cell density after 23 h of growth was approximately constant regardless of the starting conditions (Supplementary Figures S4–S6), the only parameter that changed from day-to-day was the fraction of resistant cells. To examine how the final resistant fraction depended on the initial resistant fraction on a given day, we used the time course data (Figure 1A) to generate a 'difference equation' map (Figure 1B) relating the fraction of resistant cells at the end and beginning of each day. As expected, the difference equation is non-monotonic as a result of the 'overshoot' discussed previously, and the equilibrium fraction can be obtained by finding where the difference equation map crosses the 45-degree line. In principle, if the underlying difference equation is known, then one can estimate the dynamics of the population over time by repeated application of the difference equation (or by the process of cobwebbing illustrated in Figure 1B).

In an attempt to map the difference equation using data from a single day (instead of the 8-day time course used in Figure 1A and B), we started cultures at a range of different initial resistant fractions and measured the resulting final resistant fractions after a single day of growth (Figure 1C). Such maps obtained over a single day of growth recapitulated the dynamics observed over multiple days, but with a slight overestimate of the equilibrium-resistant fraction (Figure 1B; Supplementary Figure S7). As might be expected, cultures grown at higher antibiotic concentrations had a larger equilibrium fraction of resistant cells (Figure 1C). However, the difference equations revealed that over a broad range of conditions, the sensitive cells could invade when present at low frequency. Starting with a resistant fraction below the equilibrium leads to an initial overshoot in the fraction of resistant cells in the population. After the overshoot, the resistant fraction proceeds to evolve to the equilibrium fraction, which is independent of the initial composition of the population. The resistant cells are not driven extinct by the sensitive 'cheater' cells because β-lactamase is largely contained within the periplasmic space of the resistant cells (Nikaido and Normark, 1987; Livermore, 1995; Dugatkin *et al*, 2003), thereby giving them some preferential access to the benefits of their 'cooperative' behavior (Gore *et al*, 2009). Since both resistant and sensitive cells can invade the population when present at low frequency, we observe coexistence of the two strains even in our well-mixed liquid cultures (Nowak and Sigmund, 2004; Doebeli and Hauert, 2005; Dugatkin *et al*, 2005; Gore *et al*, 2009). This coexistence between 'cooperators' and 'cheaters' is similar to what is

observed when individuals are playing the cooperative 'snowdrift' game (Gore *et al*, 2009), although it is important to note that our experimentally observed overshoot in resistant fraction over time (Figure 1) indicates that the interactions between different cell types here are much richer than are assumed in the standard models in game theory.

## A simple model captures the population dynamics

To better understand the population dynamics, we developed a simple model that describes the growth of the bacteria in the presence of antibiotics (Figure 2A and B; Supplementary Figure S8). For the range of antibiotic concentrations we probed, the resistant cells were essentially unaffected and grew at a constant rate of $\gamma_R$ (Supplementary Figures S8, S9B, and S10). We assumed that sensitive cells grow at a rate of $\gamma_S > \gamma_R$ for antibiotic concentrations below their MIC, but die at a rate of $\gamma_D$ for higher concentrations (Supplementary Figures S3, S8, and S9). Plating experiments showed that, in addition to cell death, we should incorporate a short lag phase that follows after inoculation of the bacteria into fresh media, during which bacteria neither divide nor die (Supplementary Figure S9). We modeled antibiotic

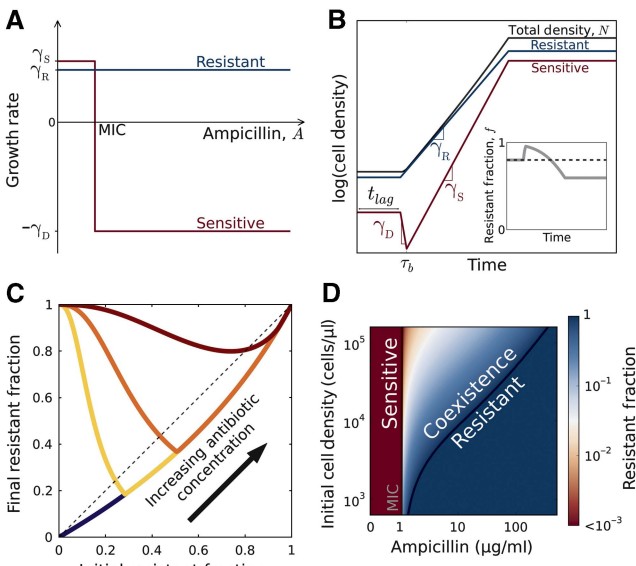

**Figure 2** A simple model describes the population dynamics of a cooperative antibiotic resistance plasmid in the β-lactam antibiotic ampicillin. (**A**) Growth rates of resistant (blue) and sensitive (red) bacteria as a function of antibiotic concentration. Free of the metabolic cost associated with resistance, sensitive cells grow faster than resistant cells ($\gamma_S > \gamma_R$) at antibiotic concentrations below the MIC of the sensitive bacteria. Above the MIC, sensitive cells die at a rate of $\gamma_D$. (**B**) The population dynamics within a single competition cycle (1 day). During the lag phase ($t < t_{lag}$), neither cell type divides nor dies, but the antibiotic is constantly hydrolyzed by resistant cells. After the lag phase, each sub-population grows at a rate that depends on the extracellular antibiotic concentration. At time $\tau_b$, the extracellular antibiotic concentration drops below the MIC of the sensitive cells. Cell growth ceases when the total population density reaches saturation. Inset: the time trace of the resistant fraction within a single day. (**C**) The model gives rise to difference equations that resemble experimental data (Figures 1C, 3A, and B). (**D**) The equilibrium-resistant fraction predicted by our model as a function of the antibiotic concentration and the initial cell density. According to the model, coexistence between resistant and sensitive cells is possible at antibiotic concentrations above the MIC of sensitive cells.

degradation phenomenologically using Michaelis–Menten kinetics with a maximum rate per cell $V_{max}$ and an effective Michaelis constant ($K_M$) (Supplementary information). While this model clearly neglects many aspects of bacterial growth in antibiotics, it successfully captures the key features of the dynamics (Figures 1C and 2C) and predicts conditions that enable coexistence between resistant and sensitive cells (Figure 2D).

We obtained an exact analytic solution of this model that describes the dependence of the equilibrium-resistant fraction, $f_R$, on the initial antibiotic concentration, $A_i$, and initial cell density, $N_i$. The model predicts that the equilibrium fraction scales in the following manner:

$$f_R \sim \frac{A_i + K_M \ln(A_i/\text{MIC}) - \text{MIC}}{V_{max}N_i} \overset{A_i \gg K_M, \text{MIC}}{\Rightarrow} \sim \frac{A_i}{V_{max}N_i}$$

This relationship is surprisingly insensitive to many parameters, including the length of the lag phase, rate of cell death, and cost associated with resistance (Supplementary information). In particular, our analytic solution of the model predicts that the resistant fraction at equilibrium increases approximately linearly with the antibiotic concentration, a prediction borne out in experimental difference maps obtained at multiple antibiotic concentrations (Figure 3A–C). Moreover, the model predicts that the equilibrium fraction is inversely proportional to the starting cell density. This prediction was experimentally confirmed by measuring the difference equations at four different starting cell densities. In each case, the equilibrium-resistant fraction increases linearly with antibiotic concentration, but with slopes that decrease with increasing initial cell density (Figure 3A–C). We therefore find a surprising simplicity to the population dynamics of the antibiotic resistance plasmid in the population, despite the biological complexity of the interaction between the cells and the antibiotic.

In addition to providing significant insight into the population dynamics, the model can quantitatively describe the experimental data. To acquire realistic parameters for the model, we measured the growth rate of resistant bacteria ($\gamma_R = 1.1/\text{h}$; Supplementary Figure S9) and the relative growth rate of sensitive bacteria ($\gamma_S/\gamma_R = 1.15$; Supplementary Figure S11). Together, these allowed us to deduce the overall metabolic cost of carrying the plasmid ($\gamma_S - \gamma_R = \sim 0.17/\text{h}$), which includes the cost of plasmid maintenance, of expressing the β-lactamase enzyme, and of expressing a red-fluorescent protein used for tracking the resistant fraction (Supplementary Figure S1). Control experiments using another plasmid that did not express a fluorescent protein exhibited similar population dynamics (Supplementary Figure S12). We proceeded to measure the death rate of sensitive bacteria in the presence of the antibiotic (2.8/h; Supplementary Figure S9) and the lag time before cell growth/death (1 h; Supplementary Figure S9).

Using these experimentally measured parameters, we then fit our 30 measured equilibrium fractions (in Figure 3C) to obtain estimates of MIC = 1.1 μg/ml, $V_{max} = 10^6$ molecules/(CFU·s), and $K_M = 6.7$ μg/ml. This MIC is slightly lower than our measured value ($\sim 2$ μg/ml; Supplementary Figure S3) because antibiotic concentrations below the measured MIC already partially inhibit the growth of sensitive bacteria (Supplementary Figure S3). In addition, our fitted value for

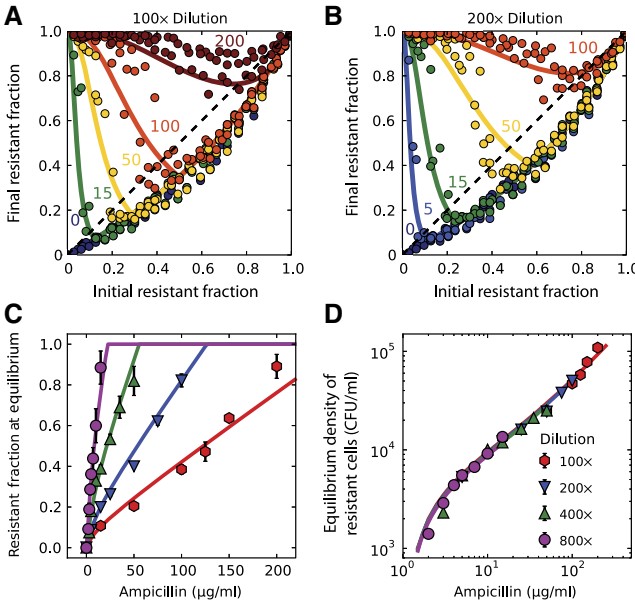

**Figure 3** Experimental difference equations confirm model predictions regarding the equilibria and dynamics of resistant and sensitive bacteria. (**A**, **B**) Experimental difference equations obtained at two dilution factors (100 × and 200 ×) and different antibiotic concentrations. At a given antibiotic concentration, an increase in the dilution ratio leads to stronger selection for resistance. Each difference equation plotted in (A, B) includes the data obtained on three different days. Measurement error from flow cytometry was typically smaller than symbol size. (**C**) The equilibrium fractions as a function of ampicillin concentration at four different dilution factors (see Supplementary Figure S13 for difference equations). The relationship is approximately linear for antibiotic concentrations higher than $K_M$. The equilibrium fractions were extracted from the difference equation plots by determining the intersection between the difference equations and the diagonal line (dashed line in A). Error bars represent standard error of the mean ($n = 3$). (**D**) Plotting the initial density of resistant cells at equilibrium as a function of antibiotic concentration reveals a data collapse that extends over two orders of magnitude in the concentration. (**A–D**) Solid curves show a single fit of the model to all the experimental data.

the maximum rate of hydrolysis per cell $V_{max}$ is reasonable since a single enzyme can hydrolyze as many as $\sim 10^3$ molecules per second (Nikaido and Normark, 1987). Although the estimate of $K_M$ agrees with the literature values (from 4.9 to 26.5 µg/ml; Livermore *et al*, 1986; Zimmermann and Rosselet, 1977; Dubus *et al*, 1994), we note that the $K_M$ in our model is a phenomenological parameter because antibiotic hydrolysis occurs both inside and outside the cells (Livermore, 1995; Zimmermann and Rosselet, 1977). The resistant fraction at equilibrium in our model increases linearly with the antibiotic concentration for $A > K_M$, but deviates slightly from linearity for $A < K_M$ due to the Michaelis–Menten kinetics of antibiotic degradation (Figure 3C). This simple model not only captures the behavior of the equilibrium fractions, but also successfully predicts the experimental difference equations using the same parameter values (Figure 3A and B; Supplementary Figure S13).

Another way to think about the scaling predicted by the model is that, at equilibrium, the number of resistant cells is proportional to the antibiotic concentration ($N_{Ri} = f_R \cdot N_i \sim A_i$). Indeed, a plot of the equilibrium density of resistant cells against the antibiotic concentration revealed a striking collapse of the data extending over two orders of magnitude in the antibiotic concentration (Figure 3D). Intuitively, more resistant

cells would be required to deactivate larger amounts of the antibiotic within a fixed period of time. Non-intuitively, the model predicts that the time necessary for a bacterial population to saturate in the presence of the antibiotic is minimized at a resistant fraction that corresponds neither to the equilibrium fraction nor to a fully resistant population (Supplementary Figure S14). Given the similarity between our experimental difference equations and the well-known 'logistic equation' from theoretical ecology (May, 1976), we used our model to characterize when the equilibrium fraction is expected to become unstable, leading to oscillations around the equilibrium. We found that the equilibrium fractions should become unstable as the antibiotic concentration decreases; however, the size of the oscillations does not become large enough to observe experimentally (Supplementary Figure S15).

## Addition of a β-lactamase inhibitor selects for resistance

Given the predictive power of the model, we explored the expected consequences of adding a β-lactamase inhibitor such as tazobactam, which is used clinically together with many β-lactam antibiotics (Bush, 1988; Livermore, 1995; Drawz and Bonomo, 2010). Tazobactam competitively binds β-lactamase enzymes (Bush, 1988; Drawz and Bonomo, 2010) and prevents them from hydrolyzing the antibiotic, leading to an increase in the effective $K_M$. A sufficiently large increase in the $K_M$ can significantly compromise the ability of resistant cells to degrade the antibiotic, leading to complete inhibition of bacterial growth (Supplementary Figure S16). However, if the increase in $K_M$ is not sufficiently large, then the resistant cells may survive the treatment, but the larger $K_M$ would hinder their ability to protect sensitive cells against the antibiotic. Specifically, as the equilibrium fraction of resistant cells is proportional to $K_M$, the model predicts that adding a β-lactamase inhibitor will lead to an increase in the resistant fraction. We have tested this prediction and found that the addition of tazobactam can indeed result in a completely resistant population (Figure 4A; Supplementary Figure S17).

Not only does the model provide qualitative insight, but it also makes surprisingly accurate quantitative predictions about the population dynamics that take place in the presence of the inhibitor. Although the actual mechanism of inhibition is more complicated (Bonomo and Tolmasky, 2007), we modeled tazobactam as a competitive inhibitor, which increases the $K_M$ to $K_{eff} = K_M \cdot (1 + [I]/K_I)$, where $[I]$ and $K_I$ are the inhibitor concentration and dissociation constant, respectively. As the equilibrium fraction increases linearly with $K_M$, the model predicts that it should also increase linearly with the inhibitor concentration $[I]$. To probe this predicted dependence of the equilibrium fraction on the inhibitor concentration, we measured the equilibrium fractions from maps of difference equations obtained at varying tazobactam concentrations (Figure 4B and C). We successfully fit the new 31 equilibrium fractions (Figure 4C) using one additional free parameter $K_I$, confirming the predicted linear dependence on the inhibitor concentration. The $K_I$ from the fit (4.6 ng/ml) was well within the literature values (3–11.4 ng/ml; Bret *et al*, 1997; Kitzis *et al*, 1988; Drawz and Bonomo, 2010). Remarkably, using the value

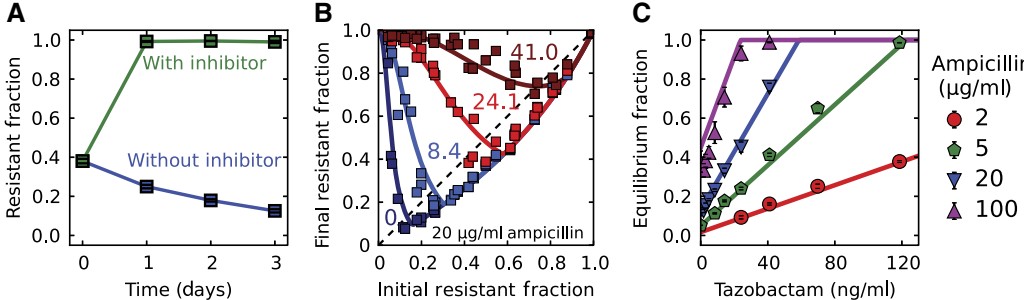

**Figure 4** As predicted by the model, addition of the β-lactamase inhibitor tazobactam increases the fraction of resistant cells in the population. (**A**) Sensitive *E. coli* cells increase in frequency when grown in 20 μg/ml ampicillin in the absence of tazobactam; however, the addition of the inhibitor at a concentration of 1000 ng/ml results in a completely resistant bacterial population. Cultures were diluted daily by a factor of 100 × into fresh media containing 20 μg/ml ampicillin. Error bars represent standard error of the mean of four different bacterial cultures. (**B**) Experimental difference equation maps for four different concentrations of the inhibitor tazobactam (in ng/ml) at a background of 20 μg/ml ampicillin and a dilution factor of 100 × (see Supplementary Figure S17 for more difference equations). Each difference equation map contains the data obtained on three different days. (**C**) As predicted by the model, the equilibrium fractions depend linearly on the concentration of the inhibitor tazobactam with a slope that depends on the ampicillin concentration. The equilibrium fractions were extracted from the difference equation plots by determining the intersection between the difference equations and the diagonal line (dashed line in A). Error bars represent standard error of the mean ($n = 3$). (B, C) Solid curves show a fit of the model to all the experimental data with a single free parameter of $K_I = 4.6$ ng/ml (other parameters held fixed).

of $K_I$ obtained from the fits to the equilibrium fractions successfully recapitulated the dynamics across the entire range of the difference equations (Figure 4B; Supplementary Figure S17).

To verify that our conclusions were not limited to tazobactam, we tried the β-lactamase inhibitor sulbactam, which is often administered together with ampicillin clinically (Foulds, 1986; Bush, 1988; Drawz and Bonomo, 2010). We found that, at least for our experimental conditions (*E. coli* bacteria inoculated at an initial cell density of ∼$10^5$ cells/μl), the addition of sulbactam can lead to the accelerated spread of resistant bacterial cells in a range of clinically relevant antibiotic concentrations (Supplementary Figure S18).

## Discussion

We have presented a quantitative analysis of the population dynamics that stem from the cooperative nature of antibiotic inactivation, and which can lead to coexistence between sensitive cells and resistant cells. Our analysis was based on two key features: (1) the presence of a metabolic cost associated with being resistant, and (2) the inactivation of the antibiotic by resistant cells. When both features apply, our model suggests that resistant and sensitive cells may coexist at high concentrations of the antibiotic, with the fraction of resistant cells approximately proportional to the antibiotic concentration divided by the cell density. We found that this simple dependence on antibiotic concentration and cell density successfully predicts the equilibrium fraction of resistant cells over two orders of magnitude in antibiotic concentration (Figure 3D).

This model not only agrees quantitatively with experimental data, but it also provides insight into the conditions that enable coexistence between resistant and sensitive cells. For example, a recent study observed coexistence with a mutated β-lactamase enzyme that inactivated the antibiotic outside the cell (Dugatkin *et al*, 2005), allowing resistant cells to efficiently 'share' their resistance with the bacterial population to support coexistence. However, in our study, we were able to observe coexistence even with a wild-type β-lactamase enzyme, which is primarily periplasmic (Livermore, 1995). To properly interpret these results, it is important to recognize that the site of antibiotic inactivation determines the degree of preferential protection offered to resistant cells. Furthermore, as long as resistant cells are sufficiently protected to be unaffected by the antibiotic, only the overall rate of antibiotic inactivation is important in determining the dynamics between resistant and sensitive cells. Hence, even if antibiotic inactivation occurs inside the cell, it is still a cooperative behavior that may allow sensitive cells to survive.

The interplay between initial cell density and antibiotic concentration is often important in determining growth dynamics in antibiotics (Brook, 1989; Tan *et al*, 2012). Likewise, our model suggested that the key parameter in governing the population dynamics was not the antibiotic concentration, but the ratio between the antibiotic concentration and the initial cell density. Specifically, we found that at high cell densities, resistant cells could protect sensitive cells against antibiotic concentration as high as 200 μg/ml (Figure 3A), which is 100-fold higher than the MIC of sensitive cells. Given the cooperative nature of antibiotic inactivation, it is likely that other ecological factors will be important to consider when attempting to understand the evolution of antibiotic resistance (Celiker and Gore, 2012; Datta *et al*, 2013; Sanchez and Gore, 2013).

One might worry that our conclusions may be limited to laboratory strains as natural strains would be better adapted to plasmids found in the wild. However, our model and experiments argue that the equilibrium fraction depends only weakly on the fitness cost of carrying the resistance plasmid (Supplementary Figure S19). Compensatory mutations that alleviate the cost of resistance (Bouma and Lenski, 1988; Dahlberg and Chao, 2003; Andersson, 2006) will increase the time it takes the population to settle into its equilibrium fraction, but will not significantly change that fraction. Since our model only uses a few key phenotypic traits to characterize the outcome of bacterial growth in the antibiotic, it should be broadly applicable in describing both intra-species (Dugatkin *et al*, 2005) and inter-species (Perlin *et al*, 2009) dynamics.

Within the framework of our model an important qualitative difference between using a bactericidal versus a bacteriostatic antibiotic is that the overshoot of the resistant fraction above the equilibrium fraction should only appear when using a bactericidal antibiotic (Figure 1A; Supplementary Figure S20). The lower the initial resistant fraction is, the longer it takes for the antibiotic to be inactivated, and the more opportunity there is for a bactericidal antibiotic to kill the sensitive strain and promote the growth of the resistant strain.

Throughout our experiments, we limited ourselves to antibiotic concentrations that do not affect the growth of resistant cells. However, at high enough concentrations, a bactericidal antibiotic may lead to lysis of resistant cells and the subsequent release of their β-lactamase enzymes into the extracellular space (Sykes and Matthew, 1976). Since these enzymes inactivate the antibiotic even faster extracellularly, the death of resistant cells may further increase the cooperative nature of bacterial growth in the antibiotic (Tanouchi *et al*, 2012). Such a scenario may explain the observed non-monotonic selection for resistance and difference equation maps that deviate from our model at high concentrations of the β-lactam antibiotic piperacillin (Supplementary Figure S21).

Understanding how the fraction of resistant bacteria changes with time is a central goal in studying antibiotic resistance. This already difficult task is further complicated by cooperative behaviors that allow resistant microbes to 'share' their resistance with the rest of the bacterial population. The cooperative nature of antibiotic inactivation causes the fitness of resistant cells to decrease as their fraction in the bacterial population increases (i.e., it leads to negative frequency-dependent selection; Dugatkin *et al*, 2005; Figure 3A and B). Overall, this enables coexistence between resistant and sensitive cells, even in the absence of the spatial structure present in biofilms (Kerr *et al*, 2002; O'Connell *et al*, 2006; Narisawa *et al*, 2008), interactions between bacteria and antibiotic degradation products (Palmer *et al*, 2010), bacterial persistence (Lewis, 2007), and indole production (Lee *et al*, 2010). As antibiotic inactivation is a frequent mechanism of antibiotic resistance (Wright, 2005), similar population dynamics may appear with other classes of antibiotics (e.g., macrolides and aminoglycosides) and with chromosomally encoded enzymes. However, despite the potential ubiquity of cooperative antibiotic resistance, the social aspect of antibiotic resistance remains underappreciated, highlighting the importance of quantitatively characterizing social interactions to gain a thorough understanding of the maintenance of phenotypic and genotypic diversity within populations.

## Materials and methods

### Strains

All strains are derived from *Escherichia coli* DH5α. The resistant strain contained the pFPV-mCherry plasmid (Drecktrah *et al*, 2008) (also see Addgene plasmid 20956), expressing a TEM-1 β-lactamase enzyme and an mCherry fluorescent protein. In addition, the resistant and sensitive strains expressed cerulean and yellow fluorescent protein genes, respectively, under the promoter $P_{lacUV5}$, and a kanamycin-resistant gene, both carried on the plasmid pZS25O1 +11 (Lutz and Bujard, 1997; Garcia and Phillips, 2011) (origin of replication: pSC101). Control experiments in which the cerulean and yellow fluorescent

markers were swapped gave nearly identical difference equation maps (Supplementary Figure S22).

## Competition experiments

All cultures were grown in a shaker at 500 r.p.m. and 37°C. Before the competition experiments, single colonies of resistant and sensitive strains were grown separately in 5 ml of lysogeny broth (LB) together with antibiotics for selection for 23 h. The saturated cultures (corresponding to a density of ~$10^7$ cells/μl) were diluted by a factor of 100× and co-cultured at different fractions in 96-well plates containing LB and 5 μg/ml of kanamycin for another 23 h to synchronize the growth state of both strains (see Supplementary Figure S11). All competition experiments were carried out using synchronized mixed cultures. The cultures were diluted into 96-well plates containing 5 μg/ml of kanamycin, LB, and appropriate concentrations of ampicillin, tazobactam, and sulbactam, and grown for another 23 h. In multi-day experiments, cultures were serially diluted into 96-well plates containing freshly prepared media with appropriate concentrations of antibiotics. Control experiments showed that the population dynamics were similar regardless of whether kanamycin was absent or present at 5 μg/ml (Supplementary Figure S23). In addition, control experiments showed that similar growth dynamics apply in other β-lactam antibiotics (Supplementary Figure S24). Fractions were determined using flow cytometry on a BD-LSR II and confirmed by plating (Supplementary Figures S1 and S2).

## Supplementary information

## Acknowledgements

We would like to thank the personnel at the Koch Institute Flow Cytometry Core at MIT for experimental help. EAY was supported by the National Science Foundation Graduate Research Fellowship (http://www.nsfgrfp.org/) under Grant No. 0645960. HXC was supported by MIT's Undergraduate Research Opportunities Program (http://web.mit.edu/urop/). MSD was supported by the NDSEG Fellowship (http://ndseg.asee.org/). The laboratory acknowledges support from an NIH R01 (no. GM102311-01; http://www.nlm.nih.gov/), NIH R00 Pathways to Independence Award (no. GM085279-02; http://www.nlm.nih.gov/), National Science Foundation CAREER Award (no. PHY-1055154; http://www.nsf.gov/), Pew Fellowship (no. 2010-000224-007; http://www.pewtrusts.org/), Sloan Foundation Fellowship (no. BR2011-066; http://www.sloan.org/sloan-research-fellowships/), and an NIH New Innovator Award (no. DP2AG04279; http://commonfund.nih.gov/newinnovator/).

*Author contributions:* EAY, HXC, MSD, and JG designed the experiments. EAY, HXC, and MSD did the experiments. TA contributed ideas to help design early experiments. EAY, HXC, and JG analyzed the data and wrote the paper. All authors discussed the results and commented on the manuscript.

## Conflict of interest

The authors declare that they have no conflict of interest.

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

**The 2nd**
**International Conference**
**on Genomics in the**

# Americas

**September 12 - 13**
**Sacramento, USA**

Co-organizer : UC DAVIS

# The 8th

**International Conference**
**on Genomics**

**October 30 - November 1**
**Shenzhen, China**

Co-organizer : GigaScience

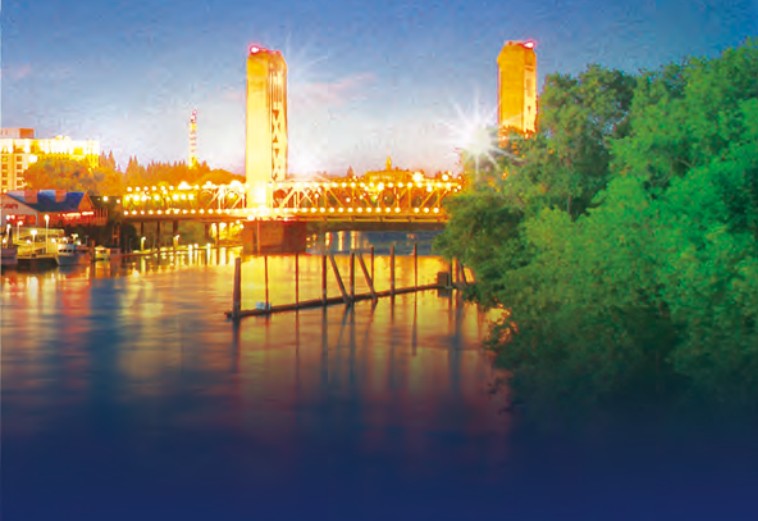

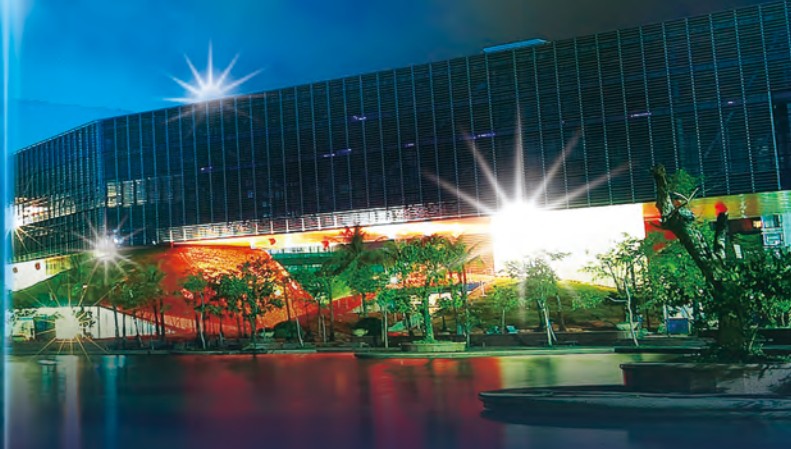

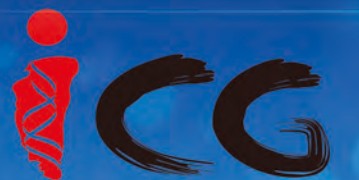

# Join Us for 2013
## International Conference on Genomics

Over the past seven years, the International Conference on Genomics (ICG) has been one of the top grade gathering of global thought leaders in genomics featuring latest advancements in genomic-related fields. This year, BGI continues to hold series ICG conferences, including ICG-8, ICG-Americas 2013, and ICG Europe 2013. These gatherings will be an excellent opportunity to exchange your research experience and latest discoveries, as well as the new insights into future development of life science.

**www.icg-2013.org**      bgi-event@service.genomics.cn      +86-755-25273340      Organizer : 华大基因 BGI