## [Review Process File · Molecular Systems Biology]

Bacterial cheating drives the population dynamics of cooperative antibiotic resistance plasmids

Eugene A. Yurtsev, Hui Xiao Chao, Manoshi San Datta, Tatiana Artemova, Jeff Gore

Corresponding author: Jeff Gore, MIT

Review timeline:

Submission date:	10 February 2013
Editorial Decision:	18 March 2013
Revision received:	08 May 2013
Editorial Decision:	04 June 2013
Revision received:	04 July 2013
Accepted:	07 July 2013

Editor: Maria Polychronidou

Transaction Report:

1st Editorial Decision

19 March 2013

Thank you again for submitting your work to Molecular Systems Biology. We have now heard back from the three referees who accepted to evaluate the study. As you will see, the referees find the topic of your study of potential interest and are supportive. However, they raise a series of concerns and make suggestions for modifications, which we would ask you to carefully address in a revision of the present work. While the additional control requested by reviewer #2 will strengthen the conclusions of the study, the further reaching experimentation suggested by this reviewer is not mandatory, even though we certainly welcome inclusion of the data if available.

Please resubmit your revised manuscript online, with a covering letter listing amendments and responses to each point raised by the referees. Please resubmit the paper ****within one month**** and ideally as soon as possible. If we do not receive the revised manuscript within this time period, the file might be closed and any subsequent resubmission would be treated as a new manuscript. Please use the Manuscript Number (above) in all correspondence.

REFeree REPORTS:

Reviewer #1:

I really enjoyed reading this paper. The work is elegantly carried out and clearly presented. It provides new insights into understanding of population-level tolerance to antibiotics by bacterial

populations. I recommend it be accepted for publication with some revisions/clarifications mostly on the technical aspects.

The setting is similar to the author's previous work (Gore et al, 2009), where the cooperative nature of an enzyme (beta-lactamase in this study and invertase in the previous study) realizes the logic of so-called snow-drift game. The asymmetry in benefit received by two strains is a critical parameter that determines the equilibrium fraction.

Remarkably, the authors were able to derive a simple relationship connecting the equilibrium resistant fraction with antibiotic concentration, dilution rate and beta-lactamase inhibitor concentration.

I find that the increased resistant fraction with increasing beta-lactamase inhibitor very interesting. While this conclusion comes naturally from the authors' analysis, it is fairly counter-intuitive from the clinical standpoint. Has this been observed in clinical/animal model studies? It is possible that the inhibitor concentrations used in this study is much lower than the clinically used level. For example, I assume that the authors need to use low enough concentration so that the resistant cells are still able to degrade ampicillin in a reasonable time period to support the overall growth. This is not critical for this paper but it will strengthen the paper.

Other specific comments:

1. The authors should note two recent publications from the You lab on related topics, and discuss them in appropriate places.

Between lines 230-233, the authors mentioned the importance between antibiotic concentration and initial cell density. This was analyzed in a different context in "Tan et al, The inoculum effect and band-pass bacterial response to periodic antibiotic treatment, *Mol Syst Biol*. 2012". Here, the phrase "...unlike previous studies..." is inaccurate.

When discussing the survival of sensitive bacteria in a population, they authors should also note "Tanouchi et al, Programming stress-induced altruistic death in engineered bacteria, *Mol Syst Biol* 2012."

2. Figure 1. Provide specific explanations of the different traces, even though they might appear self-evident. For each trace, was the initial fraction determined by mixing ratio or actually measured? Is each data point from a single measurement or average of multiple measurements, etc?

3. The fitting procedure was not clear to me. Did the authors use the data in Fig. 3C (and later Fig.4C for K_i) for the fitting?

4. Section 6 in Supplementary info where the authors simplify the expression of f_{eq} . What is "C" term? Is it $1/\tau_{R_lag}$? On a related point, the authors noted the weak dependence of f_{eq} (f_R) on the metabolic cost of resistance (Supplementary Figure 15). Was this calculated from the unsimplified equation from Section 6?

5. Fig. 4B and C. The range of inhibitor concentration does not seem to match between the two panels. For example, the final resistant fraction with 4.9 inhibitor in Fig. 4B is around 0.7 while the same fraction is achieved at around 40 in Fig. 4C (for 20 ug/ml ampicillin). Please clarify.

6. Between lines 172-175, the authors noted a non-intuitive prediction that "... the time necessary for a bacterial population to saturate in the presence of the antibiotic is minimized at a resistant bacterial population that corresponds neither to the equilibrium fraction nor a fully resistant population". On the technical level, this is indeed interesting. What's the potential significance of this prediction? And what would be an intuitive explanation for this prediction, given the model?

7. Likewise, what's the significance of oscillatory dynamics at low antibiotic concentrations (particularly when they can't be observed experimentally).

8. Description of Supplementary Figure 9 is confusing, even though the overall point is clear - that sensitive cells grow faster due to less metabolic burden.

Reviewer #2 :

Referee report on the paper: "Bacterial cheating drives the population dynamics of cooperative antibiotic resistance plasmids" by Yurtsev et al.

Short summary of the paper:

The authors analyze the social interaction between resistant and not-resistant bacteria when resistance is based on the periplasmic (or cytosolic) degradation of the antibiotics. They study it using coli as a model, ampicillin as the antibiotics and a plasmid-borne TEM-1 gene as the resistance mechanism. Starting with various initial fractions of resistance to non-resistance, the authors characterize the change in this fraction to the next day for a protocol of daily dilution into fresh medium with a defined level of antibiotics. This allows the authors to draw recurrence maps of frequency changes for different levels of antibiotics that allow the accurate prediction of long-term behavior of the antibiotics. They find that the population reaches a co-existence as a steady-state. The co-existence fraction of resistant cells depends on the level of the antibiotics. Strikingly, they find that in their setup, the recurrence map is not monotonous but that for low frequencies of resistant cells, their frequency at the next day increases dramatically and subsequently falls off slowly to the steady-state frequency.

The details of the systems dynamics can be captured by a very simple phenomenological model which assumes that resistant cells are completely insensitive to the drug and degrade it with simple michaelis-manten kinetics, while sensitive cells die at a constant rate if drug levels are above the MIC and grow at a constant rate when drug levels are below the MIC. Importantly, sensitive cells' growth-rate is higher than resistant cells, as they save the cost of resistant (plasmid growth + Ab production and action). The most important finding is that steady-state co-existence frequency is inversely-dependent on inoculum size and linearly dependent on drug concentration. Most other parameters of the population only contribute marginally to this relation under relevant parameter assumption. The full dynamics, however is dependent on parameters in a more complex manner. The authors go-on and measure independently all the easily measured parameters of their model which allow them to very accurately fit other constants of this system (such as the resistance michaelis manten parameters). They find that this extraction is consistent between experiments and is in good agreement with biochemical measurements.

Finally, the authors nicely demonstrate how, in their system, the addition of a b-lactamase inhibitor (tazobactam) lead to a counter-intuitive behavior - co-existence is eliminated leading to the fixation of the resistant strain with no harmful impact on its levels. The affinity of the inhibitor to the b-lactamase can be extracted from experiments and is in good agreement with biochemical data.

-

Referee comments

General

I think that this paper addresses very elegantly the well observed fact of partial rescue of non-resistant strains by resistant ones. It nicely demonstrates the ability to use simple phenomenological models to address this problem and shows the counter-intuitive impact of social interactions on population structure. As the authors stress, the existence of satellite colonies is well known to any student and a previous paper (the Dugatkin paper he authors cite) have demonstrated co-existence in liquid media. However, this paper extends beyond these observations and I think that it contributes nicely to our understanding of the social impact of antibiotics resistance and is worthwhile

publishing in MSB with minor revisions.

Specific comments

 There is one control experiment I was missing and another set of measurement which I think would add to the completeness of the work. The rest of my comments regard the discussion and representation of results.

Additional experiments

 1. Growth rate of resistant strains. The authors assume in their model that the growth rate of the resistant strain is antibiotics independent in the experimental range of antibiotics. In the same manner as Figure S3, it would be worthwhile seeing the change in initial growth-rate of the resistant strain as a function of antibiotic concentration. Actually, it would be interesting to extend the analysis (in discussion) to levels of antibiotics where growth of the resistant strain is initially reduced (that is, close to the MIC of the resistant strain) - how would the frequency of resistant strain change in that regime?

1a. Can the authors show the intra-day dynamics for one set of parameters. That is - measure the abundance of resistant and sensitive population (using a flow cytometer) as a function of time every hour or so, to show that the assumed dynamics in their model (sensitive first killed than grow fast while resistant grow at same rate throughout) actually fits qualitatively with the experimental results. Probably they already did this simple measurement but just did not bother to add it to the manuscript.

2. Type of antibiotics: the authors controlled for different plasmids to show they get the same behavior. They can also control for change in antibiotics by replacing ampicillin with Carbenicillin, which is supposed to be more resistant to degradation by β -lactamase (please correct me if I'm wrong). Can their experiments show the change in parameters of the two antibiotics?

Other notes

 3. Where is the snowdrift? The authors have used extensively this term in the past to describe a similar type of social interaction. Probably this term should be mentioned in the text at least once?

4. I was missing some discussion on the impact of this interaction in bactericidal drugs vs. bacteriostatic ones. It simply means discussing the change in the γ parameter in the model.

5. Can the authors estimate or discuss the cost of carrying a plasmid vs. the cost of resistance itself? This can actually be measured fairly easy by inserting the TEM-1 gene on the fluorescent carrying plasmid or on the chromosome, but I'm not sure this is important enough to strictly require it as an additional experiment.

6. Structured population: cooperative behavior is generally favored in structured population where there is some relatedness between cells. This, however, was put into question for snowdrift type of interactions in the past (Hauert, 2004). Can the authors comment on this point in the framework of their system. This of course may depend on the nature of the structure population but some (Chuang et al, 2009, Griffin 2004, Greig 2004) has been devised in the past for microbial populations and can be used as a reference.

7. Eagle effect: recently, the Yu lab suggested that the eagle effect (where number of bacteria rise with the increased antibiotics) might be attributed to social interactions. I think that also in the case presented here one would see a similar result - the average growth rate of the population at co-existence frequency should rise as a function of antibiotics levels since the fraction of resistant cells is increasing.

Minor corrections

 1. Figure S15 - y-axis should be in log-scale.

2. Definition of parameters table in modeling supp - ff should be final frequency.
3. In supp info line 142, r should be redefined (I think) $r = \log[(N_{sf}/N_{si})/(N_{rf}/N_{ri})]$ instead of ratio of logs.
4. DUGATKIN et al reference is written in all capital letters.

Reviewer #3:

The present paper analyses the social evolutionary dynamics of antibiotic resistance. This is an important and timely topic. Although antibiotic resistance is well recognized to be fundamentally an evolutionary process, there is increasing evidence that it is also a collective process where less resistant individuals are maintained thanks to the actions of other, more resistant individuals, within the same population. The realization that bacterial antibiotic resistance can be a social process promises to be an important paradigm shift, but has not received due attention.

This thorough paper introduces a simple experimental model, quantitative experiments and mathematical modeling to investigate the frequency of a beta-lactamase in a population of *E. coli* undergoing ampicillin treatment. The beta-lactamase gene (TEM-1) is carried by a plasmid (pFPV-mCherry plasmid). The paper shows that the carrying the beta-lactamase plasmid is a cooperative trait since the deactivation of ampicillin helps non-carrying cells within the same population. Yet, carrying the plasmid has a cost and this imposes non-trivial frequency dependent selection on the antibiotic resistance.

I am very impressed with the depth of the paper. Obvious care was taken in preparing the figures and the explanations are very clear. I definitely recommend publication.

The frequency dependent selection is illustrated here very beautifully with "difference equation maps" constructed experimentally. The maps reveal a stable fraction of antibiotic resistance bacteria where the experimental data cross the bisector, explaining why mixed populations of resistant and sensitive bacteria converge to the same fraction.

An important part of this paper is the simple yet effective mathematical model. As the paper rightly puts it "the model not only agrees quantitatively with experimental data, but it also provides insight into the conditions that enable coexistence between resistant and sensitive cells". The model allows finding the relation between equilibrium density of resistant cells and antibiotic concentration which displays an impressive collapse. The prediction that reducing beta-lactamase K_M (which was tested experimentally using a beta-lactamase inhibitor) is intriguing and insightful.

Minor points:

- Navigation in the main text could be significantly improved by separating in to section and adding section headings.
- A qualitative and intuitive explanation of why decreasing K_M (and adding beta-lactamase inhibitor). This is a central point and there should be great care in making it more accessible for a broader audience.

We thank you and the reviewers for careful reading of our manuscript. Below we respond to all points raised by the reviewers and describe how we have modified the manuscript to address specific points raised. Reviewer comments are italicized and our response is in normal text. In addition, we highlighted these changes directly in the manuscript.

Reviewer #1 (Remarks to the Author):

(1) I find that the increased resistant fraction with increasing beta-lactamase inhibitor very interesting. While this conclusion comes naturally from the authors' analysis, it is fairly counter-intuitive from the clinical standpoint. Has this been observed in clinical/animal model studies? It is possible that the inhibitor concentrations used in this study is much lower than the clinically used level. For example, I assume that the authors need to use low enough concentration so that the resistant cells are still able to degrade ampicillin in a reasonable time period to support the overall growth. This is not critical for this paper but it will strengthen the paper.

We should have presented this result more clearly in the paper, especially since it may seem to contradict the rationale of using an antibiotic together with an inhibitor. In using an antibiotic/inhibitor combination, one hopes to significantly compromise the ability of resistant cells to degrade the antibiotic, thereby killing the resistant cells and clearing the bacterial infection. Indeed, at higher concentrations of the beta-lactamase inhibitor we find that the population cannot grow (Supp. Figure 15).

Clinically, ampicillin is often combined together with the beta-lactamase inhibitor sulbactam. For this reason, most studies focus on the ampicillin/sulbactam combination rather than on the ampicillin/tazobactam combination, making it difficult to find information about serum concentrations of ampicillin/tazobactam. To make a comparison to clinically relevant concentrations, we added supplementary experiments with ampicillin/sulbactam (Supp. Figure 22).

Depending on how the drugs are administered, the peak serum concentration of ampicillin ranges between 40 ug/mL – 150 ug/mL while that of sulbactam ranges between 10 ug/mL – 120 ug/mL. In our experiments, we find that at the lower concentration ranges, resistant cells are able to survive and spread quickly through the bacterial population while, at the higher concentration ranges, the growth of the bacterial population is inhibited (Supp. Figure 17).

However, we want to stress that the minimum inhibitory concentrations of ampicillin/sulbactam is expected to change with the microorganism and initial cell density of the bacterial population. In many cases, the combination of ampicillin and sulbactam at clinically relevant concentrations is known to cure the bacterial infection.

We are not aware of an equivalent phenomenon having been observed in the clinic or in animal model studies.

We included this discussion in the supplementary materials and have added the following to the

main text:

“A sufficiently large increase in the Michaelis constant (K_M) can significantly compromise the ability of resistant cells to degrade the antibiotic, leading to complete inhibition of bacterial growth (Supplementary Figure 15). However, if the increase in K_M is not sufficiently large, the resistant cells may survive the treatment, but the larger K_M would hinder their ability to protect sensitive cells against the antibiotic.”

And also:

*To verify that our conclusions were not limited to tazobactam, we tried the β -lactamase inhibitor sulbactam, which is often administered together with ampicillin clinically (Bush, 1988; Foulds, 1986; Drawz & Bonomo, 2010). We found that at least for our experimental conditions (*E. coli* bacteria inoculated at an initial cell density $\sim 10^5$ cells/ μ L), the addition of sulbactam can lead to the accelerated spread of resistant bacterial cells in a range of clinically relevant antibiotic concentrations (Supplementary Figure 17).*

Other specific comments:

1. The authors should note two recent publications from the You lab on related topics, and discuss them in appropriate places.

Between lines 230-233, the authors mentioned the importance between antibiotic concentration and initial cell density. This was analyzed in a different context in "Tan et al, The inoculum effect and band-pass bacterial response to periodic antibiotic treatment, Mol Syst Biol. 2012". Here, the phrase "...unlike previous studies..." is inaccurate.

When discussing the survival of sensitive bacteria in a population, they authors should also note "Tanouchi et al, Programming stress-induced altruistic death in engineered bacteria, Mol Syst Biol 2012."

We thank the referees for pointing out these omissions. We have added citations to both publications in appropriate places in the text.

2. Figure 1. Provide specific explanations of the different traces, even though they might appear self-evident. For each trace, was the initial fraction determined by mixing ratio or actually measured? Is each data point from a single measurement or average of multiple measurements, etc?

All three traces originate from a single resistant and a single sensitive colony that were mixed to yield three cultures each with a different initial fraction of resistant cells. Then, these three cultures were grown for one day in the absence of ampicillin to ensure that both resistant and sensitive cells experience the same environment prior to measuring their dynamics in ampicillin. After the three cultures reached saturation, we measured the fraction of resistant cells using flow cytometry, calling this fraction Day 0. These three cultures were then used as the starting point for the experiment that measured the fraction of resistant cells in ampicillin as a function of time (i.e., diluting the saturated cultures into fresh medium supplemented antibiotic every 24 hours and measuring the fraction of resistant cells in the saturated cultures.)

We updated the legend of Figure 1 to help clarify what we did experimentally.

The explanation for subplot A of Figure 1 now reads:

“(A) Experimental time traces showing the evolutionary dynamics between sensitive E. coli and an isogenic strain that is resistant as the result of a plasmid containing a β -lactamase gene. A single resistant and a single sensitive colony were used to create 3 cultures with a different initial fraction of resistant cells. These 3 cultures were then grown for one day in the absence of ampicillin to make sure that resistant and sensitive cells experienced the same growth conditions (see Materials and Methods). Then, every 23 hours, the fraction of resistant cells was measured using flow cytometry, and the cultures were diluted by a factor of 100x into fresh media containing 100 μ g/mL ampicillin. Each data point represents a single flow cytometry measurement.”

3. The fitting procedure was not clear to me. Did the authors use the data in Fig. 3C (and later Fig. 4C for K_i) for the fitting?

This interpretation is correct.

A. 3 free parameters (MIC, V_{max} , K_m) were used to fit the data set shown in Figure 3C.

We used the values acquired from this fit in conjunction with experimentally measured parameters (growth rates, death rate, time lag) to produce the model's prediction for the difference equation maps (Figures 3A and 3B) by numerically integrating a model based on differential equations (Supplementary Materials).

B. To fit the data in Figure 4C, we used a single free parameter (K_i). (Other parameter values were held fixed. The values of MIC, V_{max} and K_m were set to those acquired in the fit of Figure 3C while the growth rates, death rate and lag time were set to experimentally measured values.)

Using a single value of K_i we were able to capture not only the equilibrium fractions (Figure 4C), but also the entire difference equation maps (Figure 4B, Supplementary Figure 14).

4. Section 6 in Supplementary info where the authors simplify the expression of f_{eq} . What is "C" term? Is it $1/\tau_{R_lag}$? On a related point, the authors noted the weak dependence of f_{eq} (f_R) on the metabolic cost of resistance (Supplementary Figure 15). Was this calculated from the un-simplified equation from Section 6?

$C = 1 / (\tau_{lag} + 1/g_r * ((N_{sat}/N_i)^\alpha - 1))$, where $\alpha = (g_s - g_r)/(g_s + g_d)$.

C becomes equivalent to $1/\tau_{lag}$ when $\alpha = 0$. The α we calculate for our experiment is ~ 0.04 , so it is slightly lower than $1/\tau_{lag}$.

To understand the qualitative behavior of the equilibrium fraction (f_{eq}), it is sufficient to keep track of the inverse dependence of the equilibrium fraction (f_{eq}) on the initial cell density (N_i). It is not necessary to worry about C as it is approximately a constant. (C changes by less than 10% when the dilution factor changes from 100x to 800x.)

We updated section 06 in the supplementary materials to include these clarifications.

5. *Fig. 4B and C. The range of inhibitor concentration does not seem to match between the two panels. For example, the final resistant fraction with 4.9 inhibitor in Fig. 4B is around 0.7 while the same fraction is achieved at around 40 in Fig. 4C (for 20 ug/ml ampicillin). Please clarify.*

We thank the reviewer for catching this mistake. Figure 4B was mislabeled.

6. *Between lines 172-175, the authors noted a non-intuitive prediction that "... the time necessary for a bacterial population to saturate in the presence of the antibiotic is minimized at a resistant bacterial population that corresponds neither to the equilibrium fraction nor a fully resistant population". On the technical level, this is indeed interesting. What's the potential significance of this prediction? And what would be an intuitive explanation for this prediction, given the model?*

One might naturally argue that the population growth rate should be maximized when the population consists entirely of resistant cells because in this case the antibiotic is inactivated in the smallest period of time. Alternatively, one may argue that populations should evolve to grow fastest at their equilibrium fractions. We thought it important to state our model's prediction in the paper since the prediction does not correspond to either of these two "intuitive" explanations.

Based on the model, one would qualitatively expect the fraction of resistant cells that maximizes population growth to be larger than the equilibrium fraction:

- (1) When starting below the equilibrium fraction, sensitive cells are killed by the antibiotic. Because sensitive cells die at a faster rate than resistant cells divide, the population density may actually drop during the first few hours of growth, meaning that the population would require a longer time to reach a given cell density.
- (2) When starting above the equilibrium fraction, the antibiotic is inactivated quickly, allowing sensitive cells to grow. Since sensitive cells divide faster than resistant cells (in the absence of antibiotic), the population reaches a given density faster the more sensitive cells are present immediately following complete antibiotic inactivation.

Quantitatively, the actual value of the fraction of resistant cells that maximizes the growth rate of the population depends on the parameters; e.g., changing the growth rate of either strain can change this fraction.

7. *Likewise, what's the significance of oscillatory dynamics at low antibiotic concentrations (particularly when they can't be observed experimentally).*

We mention oscillations in the main text for the benefit of readers familiar with the logistic equation who will expect a discussion about limit cycles / chaos.

In addition, we think it is important to state explicitly that the equilibrium fraction do not have to be stable fixed points.

8. *Description of Supplementary Figure 9 is confusing, even though the overall point is clear - that sensitive cells grow faster due to less metabolic burden.*

The description is probably confusing because we were trying to make two distinct points using this figure.

As the referee correctly states, the first point is that sensitive cells grow faster than resistant cells in the absence of the antibiotic because of the metabolic cost associated with resistance.

The second point we make is that to measure the relative fitness accurately, one should co-culture sensitive and resistant strains together for a day before measuring the relative fitness. Random difference in the growth histories of the two strains (when grown separately in 5 mL cultures) lead to large variations in the relative fitness during the first day of co-growth. The first day of co-growth would correspond to Day -1 if it were plotted on Figure 1.

For this reason, in all of our experiments we co-culture the strains for a day prior to measuring difference equations maps.

We rewrote the figure caption in Supplementary Figure 9 in attempt to make this discussion clearer.

Reviewer #2 (Remarks to the Author):

I think that this paper addresses very elegantly the well observed fact of partial rescue of non-resistant strains by resistant ones. It nicely demonstrates the ability to use simple phenomenological models to address this problem and shows the counter-intuitive impact of social interactions on population structure. As the authors stress, the existence of satellite colonies is well known to any student and a previous paper (the Dugatkin paper he authors cite) have demonstrated co-existence in liquid media. However, this paper extends beyond these observations and I think that it contributes nicely to our understanding of the social impact of antibiotics resistance and is worthwhile publishing in MSB with minor revisions.

Specific comments

There is one control experiment I was missing and another set of measurement which I think would add to the completeness of the work. The rest of my comments regard the discussion and representation of results.

Additional experiments

1. Growth rate of resistant strains. The authors assume in their model that the growth rate of the resistant strain is antibiotics independent in the experimental range of antibiotics. In the same manner as Figure S3, it would be worthwhile seeing the change in initial growth-rate of the resistant strain as a function of antibiotic concentration.

The growth rate of the resistant strain starts to be affected by ampicillin only when the concentration climbs above a few thousand ug/mL, which is about an order of magnitude below the MIC of the resistant strain in ampicillin.

We included data we had previously acquired for an equivalent strain (identical ampicillin resistance plasmid, but the kanamycin resistance plasmid encodes a yellow fluorescent protein instead of a cyan fluorescent protein) showing that at a few thousand ug/mL of ampicillin the resistant strain's growth rate begins to decrease (Supplementary Figure 9).

** Actually, it would be interesting to extend the analysis (in discussion) to levels of antibiotics where growth of the resistant strain is initially reduced (that is, close to the MIC of the resistant strain) - how would the frequency of resistant strain change in that regime?*

Within the framework of our model, we do not expect to see qualitatively new features appear at antibiotic concentrations close to the MIC of the resistant cells. The population will either grow to saturation with mostly/only resistant cells for concentrations right below the MIC or else it will go extinct for concentrations above the MIC.

However, experimentally qualitatively new behavior can indeed appear close to the MIC of the resistant strain. Following the referee's suggestion, we now include difference equation map measured in the antibiotic piperacillin (Supplementary Figure 20). Here, the difference equations look as expected at low concentration of the antibiotic piperacillin, but deviate from expected

behavior at the higher antibiotic concentrations, where selection for resistance is reduced at low initial fractions of resistance cells. This data exhibits non-monotonic selection for resistance with increasing concentrations of the antibiotic.

One possible explanation for this behavior is that the higher concentration of piperacillin cause cell lysis of resistant cells and subsequent release of their beta-lactamase enzymes into the extra-cellular medium. The extra-cellular enzymes hydrolyze the antibiotic more efficiently (because there is more substrate), leading to the survival of more sensitive cells.

This discussion is now included in the main text:

“Throughout our experiments, we limited ourselves to antibiotic concentrations which do not affect the growth of resistant cells. However, at high enough concentrations, a bactericidal antibiotic may lead to lysis of resistant cells and the subsequent release of their beta-lactamase enzymes into the extra-cellular space (Sykes & Matthew, 1976). Since these enzymes inactivate the antibiotic even faster extracellularly, the death of resistant cells may further increase the cooperative nature of bacterial growth in the antibiotic (Tanouchi et al, 2012). Such a scenario may explain the observed non-monotonic selection for resistance and difference equation maps that deviate from our model at high concentrations of the β -lactam antibiotic piperacillin (Supplementary Figure 20).”

1a. Can the authors show the intra-day dynamics for one set of parameters. That is - measure the abundance of resistant and sensitive population (using a flow cytometer) as a function of time every hour or so, to show that the assumed dynamics in their model (sensitive first killed than grow fast while resistant grow at same rate throughout) actually fits qualitatively with the experimental results. Probably they already did this simple measurement but just did not bother to add it to the manuscript.

This would have been a very nice experiment to add to our manuscript to complement our results. Unfortunately, we did not manage to complete this experiment on time.

However, we would be very surprised if the basic population dynamics assumed by the model turned out to be incorrect for the following reasons:

1. We monitored the behaviors of the individual strains in the presence and absence of ampicillin in plating experiments (Supplementary Figure 8), so we know all the elements we incorporated into the differential equations are present (cell death, lag time, growth).
2. We know that the sensitive cells grow faster than resistant cells (Supplementary Figure 9).
3. The model is minimalistic in the sense that removing any of its parts produces qualitatively different behavior than what is observed experimentally. (The only exception is the lag time. Removing it does not lead to significant qualitative changes, but it does produce unreasonably high numbers for the hydrolysis rate.)

2. *Type of antibiotics: the authors controlled for different plasmids to show they get the same behavior. They can also control for change in antibiotics by replacing ampicillin with Carbenicillin, which is supposed to be more resistant to degradation by b-lactamase (please correct me if I'm wrong). Can their experiments show the change in parameters of the two antibiotics?*

We measured difference equation maps with a few other beta-lactam antibiotics (Supplementary Figure 23). As predicted by the referee, the difference equation maps reveal that a given concentration of carbenicillin selects for more resistant cells than the same concentration of ampicillin.

Here, the model makes yet another strong quantitative prediction. The slope of the equilibrium fraction vs. antibiotic for high antibiotic concentrations is proportional to the rate of antibiotic hydrolysis. This means that these slopes in the antibiotics ampicillin, penicillin G and piperacillin should be approximately the same (similar k_{cat} s for all antibiotics). While the slope in carbenicillin should be 10x larger than for the other antibiotics (the k_{cat} is 10x lower). Our experimental data shows that the slope in carbenicillin is indeed larger than in the other antibiotics by close to a factor of 10 (Supplementary Figure 23D).

Other notes

3. *Where is the snowdrift? The authors have used extensively this term in the past to describe a similar type of social interaction. Probably this term should be mentioned in the text at least once?*

The following comment was added to the main text:

“This coexistence between “cooperators” and “cheaters” is similar to what is observed when individuals are playing the cooperative “snowdrift” game (Gore et al, 2009), although it is important to note that our experimentally observed overshoot in resistant fraction over time (Figure 1) indicates that the interactions between different cell types here is much richer than is assumed in the standard models in game theory.”

4. *I was missing some discussion on the impact of this interaction in bactericidal drugs vs. bacteriostatic ones. It simply means discussing the change in the γ_D parameter in the model.*

Cell death is responsible for the large overshoot of the resistant fraction above the equilibrium fraction when starting at a low initial fraction of resistant cells. (e.g., trace that starts below

equilibrium fraction in Figure 1A). This overshoot is not expected for bacteriostatic antibiotics (see simulation in Supplementary Figure 19). Basically, the lower the initial resistant fraction is, the longer it takes for the antibiotic to be inactivated, and the more opportunity there is for a bactericidal antibiotic to kill the sensitive strain and promote the growth of the resistant strain.

We added the following paragraph to the main text:

“Within the framework of our model an important qualitative difference between using a bactericidal vs. a bacteriostatic antibiotic is that the overshoot of the resistant fraction above the equilibrium fraction should only appear when using a bactericidal antibiotic (Figure 1A, Supplementary Figure 18). The lower the initial resistant fraction is, the longer it takes for the antibiotic to be inactivated, and the more opportunity there is for a bactericidal antibiotic to kill the sensitive strain and promote the growth of the resistant strain.”

5. Can the authors estimate or discuss the cost of carrying a plasmid vs. the cost of resistance itself? This can actually be measured fairly easily by inserting the TEM-1 gene on the fluorescent carrying plasmid or on the chromosome, but I'm not sure this is important enough to strictly require it as an additional experiment.

We did a somewhat similar experiment to the one proposed in which we co-cultured two different resistant bacterial strains. The difference between the two strains was a promoter mutation that caused one of the strains to double its production of the beta-lactamase enzyme. We found that doubling the production of the enzyme hardly affected the fitness of the mutant strain (it was only lower by ~1%). Hence, we expect that most of the cost of resistance is due to the metabolic burden associated with maintaining the plasmid.

We chose not to include this discussion in the main text since disentangling the various contributions to the metabolic cost of resistance does not help to explain the population dynamics, which are determined only by the presence of an overall metabolic cost (i.e., even its actual value is not that important).

6. Structured population: cooperative behavior is generally favored in structured population where there is some relatedness between cells. This, however, was put into question for snowdrift type of interactions in the past (Hauert, 2004). Can the authors comment on this point in the framework of their system. This of course may depend on the nature of the structure population but some (Chuang et al, 2009, Griffin 2004, Greig 2004) has been devised in the past for microbial populations and can be used as a reference.

The referee has made a nice summary of the literature on this subject. Indeed, in unpublished work we have probed the effect of spatial structure on the evolution of cooperation in our sucrose metabolism system, which is closer to a “standard” snowdrift game than is the current system. In those experiments, we found experimentally that spatial structure favored cooperation. Simulations indicated that the effect described by Hauert et al depends upon the nature of population “updating”, and in particular may not apply to most situations with cells. We believe that these issues are interesting, but any discussion of these effects in the current paper would

likely be more confusing than enlightening.

7. Eagle effect: recently, the Yu lab suggested that the eagle effect (where number of bacteria rise with the increased antibiotics) might be attributed to social interactions. I think that also in the case presented here one would see a similar result - the average growth rate of the population at co-existence frequency should rise as a function of antibiotics levels since the fraction of resistant cells is increasing.

Within the framework of the model we would expect the opposite behavior. It is probably easiest to see by considering two different extremes.

In the absence of any antibiotic, the population will consist entirely of sensitive cells at equilibrium. Hence, the population will grow at a rate γ_S (the growth rate of sensitive cells).

In high concentrations of the antibiotic, the population will consist entirely of resistant cells at equilibrium. Hence, the population will grow at a rate γ_R (the growth rate of resistant cells).

Since sensitive cells divide faster than resistant cells ($\gamma_S > \gamma_R$), the population will grow faster in the absence of any antibiotic than in the presence of a lot of antibiotic.

At intermediate concentrations of the antibiotic, the growth rate should be at an intermediate value between the two growth rates.

Of course, incorporating additional features into the model (e.g., cell lysis of resistant cells and the release of beta-lactamase enzymes into the extra-cellular space) will complicate this simple conclusion.

Minor corrections

1. Figure S15 - y-axis should be in log-scale.

Changed to log-scale.

2. Definition of parameters table in modeling supp - ff should be final frequency.

Fixed.

3. In supp info line 142, r should be redefined (I think) $r = \log[(N_{sf}/N_{si})/(N_{rf}/N_{ri})]$ instead of ratio of logs.

It should be the ratio of logs.

$$N_{sf} = N_{si} * E^{(g_s * T_{sat})} \rightarrow g_s = \log(N_{sf} / N_{si}) / T_{sat}$$

$$N_{rf} = N_{ri} * E^{(g_r * T_{sat})} \rightarrow g_r = \log(N_{rf} / N_{ri}) / T_{sat}$$

Since the relative fitness is defined as the ratio between the two growth rates (g_s and g_r), the

relative fitness is:

$$r = g_s/g_r = \log(N_{sf}/N_{si}) / \log(N_{rf}/N_{ri})$$

4. *DUGATKIN et al reference is written in all capital letters.*

Fixed.

Reviewer #3 (Remarks to the Author):

Minor points:

- Navigation in the main text could be significantly improved by separating in to section and adding section headings.

We added subsections to the results section to help the reader navigate through the text.

- A qualitative and intuitive explanation of why decreasing KM (and adding beta-lactamase inhibitor). This is a central point and there should be great care in making it more accessible for a broader audience.

We added a short explanation in the text to help communicate intuitively what is going on; i.e., the addition of the beta-lactamase inhibitor tazobactam results in slower inactivation of the antibiotic by resistant cells (lower K_m), which makes it more difficult for sensitive cells to survive (leading to a higher fraction of resistant cells).

Thank you again for submitting your work to Molecular Systems Biology. We have now heard back from the referee who accepted to evaluate the study. As you will see, the referee finds the response to the comments satisfying.

While the additional experiment suggested by the referee is not required, we would like to ask you to address the point related to the Eagle effect and make (if necessary) the requested amendments to Supplementary Figure 20.

Reviewer #2 (Remarks to the Author):

I find the answers of the referees satisfying. They have clearly added much data which extends the scope of the work. I just find the fact that they haven't been able to produce the within-day change in frequency surprising. Based on my knowledge on such experiments, this should be a very simple experiment. I agree with the authors argument that it will most likely show the expected results, but still think it will strengthen the paper. In any case, I leave it to the consideration of the editor whether he sees this experiment as important or not.

A small comment: Regarding the Eagle effect. I agree with what the authors claim about the model, but the effect can probably be seen in Supp fig 20 at high levels of the antibiotics, as the level of resistant cells is decreasing with antibiotics. If true, this might be mentioned in the caption for this figure with reference to the paper from the Yu lab.

I also find the response to the comments of referee #1,#3 satisfying.

We thank you and the reviewer for the time taken to help improve the manuscript.

There are no significant changes in the main text except for one additional citation and updated figure numbers.

In the supplementary materials, we added supplementary figure 8 in response to the request of reviewer #2 to monitor the intra-day growth of bacteria in the antibiotic.

Below we respond to all points raised by the reviewer. Reviewer comments are italicized and our response is in normal text.

Reviewer #2 (Remarks to the Author):

I find the answers of the referees satisfying. They have clearly added much data which extends the scope of the work. I just find the fact that they haven't been able to produce the within-day change in frequency surprising. Based on my knowledge on such experiments, this should be a very simple experiment. I agree with the authors argument that it will most likely show the expected results, but still think it will strengthen the paper.

We have done the experiment suggested by the referee and included it in the supplementary materials (Supplementary Figure 8). The data clearly supports the assumptions made by the model. At lower antibiotic concentrations and higher initial fractions of resistant cells, sensitive cells grow virtually unhindered by the antibiotic. In contrast, at higher antibiotic concentrations and lower initial fractions of resistant cells, the antibiotic kills a significant fraction of the sensitive cells before it is inactivated. Also consistent with our model, sensitive cells manage to resume growth after having experienced a period of cell death.

A small comment: Regarding the Eagle effect. I agree with what the authors claim about the model, but the effect can probably be seen in Supp fig 20 at high levels of the antibiotics, as the level of resistant cells is decreasing with antibiotics. If true, this might be mentioned in the caption for this figure with reference to the paper from the Yu lab.

Supp. Figure 20 (now relabeled to 21) shows that when starting from low initial fractions of resistant cells, the final fraction depends non-monotonically on the antibiotic concentration. However, the equilibrium fraction of resistant cells still increases monotonically with the antibiotic concentration. Specifically, the data does not indicate that resistant cells lyse when starting at initial fractions that correspond to the equilibrium fractions. Therefore, we would not expect to see an Eagle effect (whose origin is cell death) appear when starting at the equilibrium fraction.

For populations inoculated at low initial fractions of resistant cells, the release of additional beta-lactamase into the extracellular space due to cell lysis at increased antibiotic concentrations should increase the rate of antibiotic inactivation. However, to observe the Eagle effect, the higher rate of antibiotic inactivation must make up for the cell death experienced by the bacterial population. Because the region in parameter space where this occurs could be small, cell death does not necessarily imply that an Eagle effect is present.

It is difficult to make any arguments based on the data we have for supplementary figure 20 because we only measured the final cell density (the cell density after a day of growth), which did not exhibit any significant dependence on the initial antibiotic concentration. In addition, in unrelated experiments carried out in our lab, the growth of resistant cells was monitored continuously throughout their growth and no Eagle effect was observed in the antibiotic piperacillin.

Because the cooperative nature associated with the cell death of resistant cells is important in determining the population dynamics, the *Tanouchi et al.* paper is cited at the appropriate place in the main text.